# Development and Validation of a Risk Score for Post-Transplant Lymphoproliferative Disorders among Solid Organ Transplant Recipients

**DOI:** 10.3390/cancers14133279

**Published:** 2022-07-04

**Authors:** Quenia dos Santos, Neval Ete Wareham, Amanda Mocroft, Allan Rasmussen, Finn Gustafsson, Michael Perch, Søren Schwartz Sørensen, Oriol Manuel, Nicolas J. Müller, Jens Lundgren, Joanne Reekie

**Affiliations:** 1Centre of Excellence for Health, Immunity and Infections (CHIP), Rigshospitalet, University of Copenhagen, 2100 Copenhagen, Denmark; neval.ete.wareham@regionh.dk (N.E.W.); amanda.mocroft@regionh.dk (A.M.); jens.lundgren@regionh.dk (J.L.); joanne.reekie@regionh.dk (J.R.); 2Centre for Clinical Research, Epidemiology, Modelling and Evaluation (CREME), Institute for Global Health, University College London, London WC1E6BT, UK; 3Department of Surgical Gastroenterology, Rigshospitalet, 2100 Copenhagen, Denmark; allan.rasmussen@regionh.dk; 4Department of Cardiology, Section for Lung Transplantation, Rigshospitalet, 2100 Copenhagen, Denmark; finn.gustafsson@regionh.dk (F.G.); michael.perch@regionh.dk (M.P.); 5Department of Nephrology, Copenhagen University Hospital Rigshospitalet, 2100 Copenhagen, Denmark; soeren.schwartz.soerensen@regionh.dk; 6Infectious Diseases Service and Transplantation Center, Lausanne University Hospital and University of Lausanne, 1011 Lausanne, Switzerland; oriol.manuel@chuv.ch; 7Division of Infectious Diseases and Hospital Epidemiology, University Hospital Zurich, Swiss Transplant Cohort Study, 8091 Zürich, Switzerland; nicolas.mueller@usz.ch

**Keywords:** transplantation, EBV DNA, PTLD, SOT

## Abstract

**Simple Summary:**

Post-transplant lymphoproliferative disease (PTLD) is a well-recognized complication after transplant. We developed and validated a risk score to predict PTLD among solid organ transplant (SOT) recipients. This work presents the first risk score developed and externally validated to predict risk of PTLD among SOT recipients. PTLD is a potentially severe complication after solid organ transplantation, and the best clinical strategy would be to detect the disease at early stages and prevent progression to fulminant lymphoma or even better to predict the risk of a patient developing PTLD prior to onset of disease. Our score had a strong similar discriminatory ability in both derivation and validation cohorts.

**Abstract:**

Post-transplant lymphoproliferative disease (PTLD) is a well-recognized complication after transplant. This study aimed to develop and validate a risk score to predict PTLD among solid organ transplant (SOT) recipients. Poisson regression identified predictors of PTLD with the best fitting model selected for the risk score. The derivation cohort consisted of 2546 SOT recipients transpanted at Rigshospitalet, Copenhagen between 2004 and 2019; 57 developed PTLD. Predictors of PTLD were high-risk pre-transplant Epstein–Barr Virus (EBV), IgG donor/recipient serostatus, and current positive plasma EBV DNA, abnormal hemoglobin and C-reactive protein levels. Individuals in the high-risk group had almost 7 times higher incidence of PTLD (incidence rate ratio (IRR) 6.75; 95% CI: 4.00–11.41) compared to the low-risk group. In the validation cohort of 1611 SOT recipients from the University Hospital of Zürich, 24 developed PTLD. A similar 7 times higher risk of PTLD was observed in the high-risk group compared to the low-risk group (IRR 7.17, 95% CI: 3.05–16.82). The discriminatory ability was also similar in derivation (Harrell’s C-statistic of 0.82 95% CI (0.76–0.88) and validation (0.82, 95% CI:0.72–0.92) cohorts. The risk score had a good discriminatory ability in both cohorts and helped to identify patients with higher risk of developing PTLD.

## 1. Introduction

Post-transplant lymphoproliferative disease (PTLD) is a potentially severe complication after solid organ transplant (SOT) consisting of a group of clinically heterogeneous lymphoid disorders that vary from indolent polyclonal proliferation to aggressive lymphomas [1]. Up to 20% of SOT recipients have been reported to develop PTLD; however, estimates vary by immunosuppressive treatment and patient characteristics [2].

PTLD diagnosis can be a challenge due to nonspecific clinical presentation and differences in histopathology and immunophenotypic presentations [3]. Further, it can be difficult to distinguish PTLD from other conditions associated with lymph plasmatic infiltrations such as infection or graft rejection [4]. Response to PTLD treatments such as chemotherapy or rituximab depends on timely initiation of treatment [5]. Thus, the best clinical strategy would be to detect the disease at early stages and prevent progression to fulminant lymphoma [6] or even better to predict the risk of a patient developing PTLD prior to onset of disease [7,8].

PTLD is often preceded by Epstein–Barr virus (EBV) DNAemia, and EBV can be detected in most PTLD tumors [7,8]. Hence, the detection of EBV DNA in plasma or whole blood is considered an early sign of PTLD [6], and this biomarker is used to guide clinical management of this condition in most clinical settings [7,8]. However, not all patients with EBV DNAemia develop PTLD, and PTLD may occur without relation to EBV DNAemia. Thus, the ability of EBV DNAemia to distinguish between patients who progress to develop PTLD and those who remain free of disease has been shown to be relatively low, often leading to low positive predictive values [6]. Thus, routine screening of adult SOT recipients for EBV DNAemia is not recommended according to the 2010 guidelines by the British Committee for Standards in Haematology and British Transplant Society [9].

A recent study by our group proposed a novel model to predict PTLD with good sensitivity and specificity [6]. The study showed that specific clinical characteristics and laboratory values known to impact disease progression predicted PTLD well and had superior performance compared to using EBV DNAemia detection alone.

The aim of this study was to develop and externally validate a simple and widely applicable risk score to predict PTLD in SOT recipients. The risk score will help decision-making in clinical practice through identification of individuals at highest risk of PTLD.

## 2. Materials and Methods

### 2.1. Study Design and Participants

The derivation cohort included all SOT recipients (heart, liver, lung, kidney, and pancreas) transplanted at the University Hospital of Copenhagen, Rigshospitalet, a large tertiary transplant center between January 2004 and December 2019 and registered in the Management of Post-Transplant Infections in Collaborating Hospitals (MATCH) cohort [10].

The validation cohort included all SOT recipients undergoing solid organ transplantation (heart, liver, lung, kidney, and pancreas) in the University Hospital of Zürich, Switzerland, between January 2008 and December 2018 and enrolled in the Swiss Transplant Cohort Study (STCS) [11].

### 2.2. Primary Outcome

For both cohorts cases of definite PTLD were defined based on the WHO criteria (which require a biopsy-confirmed diagnosis) [12]. However, obtaining a biopsy is not always possible due to risk of serious adverse events related to the biopsy procedure or due to a challenging tumor site. Thus, in the derivation cohort, the PTLD definition was expanded to include “probable” and “possible” PTLD to be able to also ascertain non-biopsy confirmed PTLDs [6]. Probable PTLD is characterized by EBV DNAemia together with significant lymphadenopathy (or other end-organ disease) with the absence of another cause, while in possible PTLD there are EBV DNAemia and appropriate EBV-related symptoms, such as B-symptoms without evidence of probable or definite disease [6]. The classification of PTLD was performed by a trained clinician through review of several sources, including pathology data, lab data, death causes, and journal records [6].

### 2.3. Data Sources, Risk Factors, and Definitions

For the derivation cohort, clinical characteristics as well as biochemical and virologic data were extracted from the Centre of Excellence for Personalized Medicine for Infectious Complications in Immune Deficiency (PERSIMUNE) data repository. The data repository includes nationwide data collected prospectively as part of routine care [13].

Similarly, for the validation cohort, the data were collected for STCS prospectively as part of routine care [11]. Individual patient data on specific immunosuppressive therapies were not available; however, detailed information on the immunosuppressive schemes for both cohorts can be found in previous published articles [14,15] and as Appendix A.

Risk factors for the development of PTLD were decided a priori based on the results from a previous study published by our group [6]. They include variables assessed at the time of transplant (sex, age, transplant type, year of transplant, and EBV IgG donor (D) and recipient (R) serostatus) and time-updated risk factors (number of transplants, laboratory tests (hemoglobin, CRP, platelets, and EBV DNA)) [6].

Pre-transplant EBV IgG serostatus was spilt into the two categories of high-risk (D+/R−) and low-risk (D+/R+, D−/R+, D−/R−, and unknown) as the risk of PTLD was similar across the low-risk groups. Patients with unknown D/R serostatus were grouped together with low D/R serostatus because they had similar risks of PTLD.

EBV DNA was measured in the plasma using real-time polymerase chain reaction (PCR) [6]. The lower limit of detection was >500 copies/mL in MATCH and >200 copies/mL in STCS. Values below the lower limit of detection in the two cohorts were defined as negative and values above as positive. MATCH patients who were EBV IgG seronegative at time of transplantation were monitored with EBV PCR every 2 to 4 weeks during the first 6 months of follow-up and at the discretion of the clinicians thereafter [6]. Individuals with low/standard risk EBV IgG serostatus at time of transplant were followed at the discretion of the clinicians. In the STCS, heart and lung recipients were monitored for EBV DNA on every regular visit the first 6 months after transplant independent of EBV serostatus, thereafter with larger intervals, whereas kidney and liver recipients were monitored at the discretion of the clinicians.

Other laboratory values were classified into categories according to the reference ranges used by the Department of Biochemistry at Rigshospitalet, Copenhagen University Hospital, i.e., normal, below, or above normal ranges with normal ranges defined as hemoglobin: 7.3–9.5 mmol/L for women and 8.3–10.5 mmol/L for men; CRP: <10 mg/L; and platelets: 145–390 × 10^9^/L [16].

All time-updated laboratory tests, including EBV DNA results, were lagged by 28 days to reduce both the impact of EBV DNA testing in connection with clinical symptoms and concerns about reverse causality [6]. Patients were initially classified into categories based on the most recent laboratory tests at the time of transplant (i.e., positive, negative, or unknown). A patient could subsequently move categories once a new measurement became available, with the last observation carried forward until that point. For EBV DNA, all patients were initially classified into the unknown category until they first had an EBV PCR test. Individuals who were never tested for EBV DNA remained in the unknown category for the whole follow-up.

For other laboratory tests, unknown values were grouped together with the category that had a similar incidence rate of PTLD.

### 2.4. Model Derivation

MATCH SOT recipients were included from their date of first transplant after 2004 and followed until PTLD event (first PTLD, in case of multiples), 60 days after the last visit, death, or end of the follow-up (December 2019), whichever occurred first.

Different methods of building the PTLD risk score were tested to determine whether similar predictors of PTLD were identified. Forward selection, i.e., including the most important factors first (those with *p* value less than 0.1 in the univariate Poisson regression and/or those with the greatest change in Akaike information criterion—AIC) [17], led to the same model choice, as did backward selection.

Poisson regression was used because the response variable of interest is a count (PTLD event), while for Cox regression the response variable is the time that has elapsed between some origin and an event of interest. Furthermore, the Cox proportional hazards model, as the name suggests, assumes the hazard function is proportional in the population over time. On the other hand, Poisson regression assumes constant hazards.

After the best model was found, coefficients were scaled by dividing all coefficients by the smallest coefficient and rounding to the nearest whole number. A PTLD risk score per individual was calculated based on their measured risk factors and the assigned weight from their scaled coefficients. Quintiles of the derived PTLD risk score were used to initially divide the population into five categories of risk. The incidence of PTLD in each category was assessed and categories with similar incidence rates collapsed into one group.

### 2.5. External Model Validation

The PTLD risk score was validated in the STCS cohort. The discriminatory ability of the PTLD score model in both cohorts was assessed with Harrell’s C-statistic, and the accuracy of predictions for individual patients was assessed using the Brier score. For the Brier score, smaller scores (closer to zero) indicate better forecasts [18], and for Harrell’s C-statistic, values over 0.8 indicate a strong model [19]. The PTLD risk score performance was further evaluated by comparison of the crude incidence rates and incidence rate ratios within PTLD score groups and incidence rate ratios associated with a 1-point increase in the PTLD score.

### 2.6. Calculation of an Individual’s Risk of Developing PTLD—An Example

The risk score proposed in this study can be used at any time point after the transplant, just by entering the patient characteristics available at a specific time period. For example, the risk of developing PTLD in the next 180 days can be calculated using the formula:Prob (PTLD next t = 180 days)
1 − exp (t × (−exp (β0 + β1X1 + β2X2 + …βnXn)))
where t denotes time (in this case, the next 180 days), Xn denotes each of the risk factors, and bn denotes the parameter coefficients from the Poisson regression model. The PTLD score and a tool to calculate risk of developing PTLD will become available online where the risk can be estimated by entering the individual’s own characteristics.

The risk score is not a static model because we used time-updated analysis to account for each new available laboratory measurement of the patients. Thus, patients can change groups all the time, and it is not possible to determine an overall number of patients in the high-risk group that did not develop PTLD. However, if a specific time point is fixed, it is possible to assess the probability of developing PTLD in low- and high-risk of developing PTLD.

### 2.7. Sensitivity Analyses

To check differences in the discriminatory ability of the chosen model in the derivation cohort, the performance of the PTLD risk score was also assessed with only definite PTLD events in the derivation cohort and stratifying the follow-up into two periods, the first year post-transplant and greater than 1 year for those still under follow-up.

We also ran two additional sensitivity analysis: one excluding the unknown values (missing) for the laboratory parameters and the second retaining age group in the model, as studies have reported a higher risk of PTLD in children than in adults [1].

## 3. Results

### 3.1. Patient Characteristics

A total of 2546 SOT recipients were included in the derivation cohort with 57 PTLD events (46 definite, 7 probable, and 4 possible PTLD) during 13,026.45 PYFU (IR 4.40 per 1000 PYFU, 95% CI 3.25–5.54). Figure 1 shows the probability of all PTLD events and the probability of only definite PTLD events over time. Patient characteristics at time of transplant are found in Table 1. Few patients (n = 27) received a pancreas transplant, of which 25 patients had a combined pancreas + kidney transplant and only 2 single pancreas transplants. Thus, these patients were grouped together with kidney recipients. Compared to patients that did not develop PTLD, patients who developed PTLD had higher CRP levels and abnormal platelets (high and low) at the time of transplant (0–7 days prior to transplant). A higher proportion of patients were also in the high EBV D+/R− risk category.

Characteristics of PTLD for both cohorts are found in Table 2. A little over half of the PTLD cases in both cohorts were B cells (56.1% and 54.2% in derivation and validation cohorts respectively). In relation to Ann Arbor classification of Lymphoma, for the validation cohort most of the PTLDs were classified as stage IV (most advanced stage of lymphoma, while for the derivation cohort, the higher percentage of PTLDs with information available for this variable was found in stages I (least advanced) and IV. Further, in both cohorts the majority of definite PTLD cases were positive for EBV DNA at time of PTLD diagnosis.

The periodicity of laboratory tests varied during the follow-up, most frequently in the first year after transplant (Table 3).

### 3.2. Development of the Best Model to Predict PTLD in the Derivation Cohort

Risk factors univariably associated with PTLD included EBV IgG D/R serostatus and age group both measured at the time of transplant and the most recent EBV DNA, CRP, hemoglobin, and platelet levels. In multivariable analysis an increased risk of PTLD was significantly associated with high-risk D+R−, EBV IgG serostatus at transplant and current EBV DNAemia, lower hemoglobin, and higher CRP levels (Table 4). Additionally, transplant type was not significant in the univariable analysis but was retained in the multivariable model as it improved the fit (assessed by the AIC). Age group was subsequently dropped from the multivariable model as inclusion resulted in a higher AIC as well as other risk factors not included in the final model. Forward and backward selection methods identified the same risk factors to build the best multivariate model to predict PTLD.

Lung transplant type yielded the smallest coefficient and was used to scale the other coefficients (Table 3). Positive EBV DNAemia was the strongest predictor of PTLD and contributed most to the score (+42 points), followed by high D+R− risk EBV IgG serostatus at time of transplant (+19 points); CRP levels above normal ranges (+13 points); hemoglobin levels above normal ranges (+11 points); hemoglobin levels below the normal ranges (+10 points); heart, liver, and lung transplant (+9, +4, +1 points respectively); platelets above the normal ranges (+9 points); platelets below the normal ranges (+5 points); and unknown EBV DNA (−6 points). The score was calculated for all patients in the derivation cohort and split into quintiles. There was no significant difference in incidence rate (IR) of PTLD in quintiles 1 to 4, so they were grouped together as the “low-risk group” (score ≤ 17 points), and the fifth quintile was kept separate as the “high-risk group” (score > 17 points).

Compared with those in the low-risk group, the risk of developing PTLD was almost 7 times higher in the high-risk group (IRR 6.75, 95% confidence interval (CI) 4.00–11.41). The discriminatory ability was good with a Harrell’s C-statistic of 0.82 (0.76–0.88) (Table 5).

### 3.3. Validation Cohort

In the validation cohort, 1611 patients were included with 24 definite PTLD events during 7218.53 PYFU with an IR of 3.32 per 1000 PYFU (95% CI 1.99 to 4.65), which was comparable to the derivation cohort when only the 46 definite PTLD cases were considered (IR 3.53 per 1000 PYFU (95% CI 2.51 to 4.55)), Figure 1. Patient characteristics were similar to the derivation cohort (Table 1).

The discriminatory ability of the PTLD risk score model was also good in the validation cohort and very similar to the derivation cohort (Table 4). Patients in the high-risk group had a 7 times higher risk of PTLD compared to patients in the low-risk group (IRR 7.17, 95% CI 3.05–16.82). The IRR associated with a 1-point increase in the PTLD risk score was also similar to the derivation cohort as well as the accuracy of predictions for individual patients measured by the Brier score (Table 4).

### 3.4. Calculation of an Individual’s PTLD Risk Score

The exact and the scaled coefficients used in the risk score are presented in Table 3. We will illustrate two different scenarios as examples. First, a patient who has a low-risk pretransplant EBV serostatus (+0), who was transplanted with a heart (+9), who has a positive EBV DNA (+42), who has hemoglobin levels below the normal ranges (+10), and who has C-protein levels above the normal range (+13) and platelets at the normal ranges (+0), would have a risk score of 74 points, placing them in the high-risk category (>17 points). Using the exact coefficients, this same patient would have a risk for developing PTLD in the next 180 days of 8.73%.

Secondly, if the same patient had the same characteristics above except with a negative EBV DNA (+0), normal CRP levels (+0), and normal hemoglobin levels (+0), the patient would have a risk score of 9 points, placing them in the low-risk category (≤17 points) and would have only a 0.1% risk of developing PTLD in the next 180 days.

In relation to the probability of PTLD, we fixed two specific time points (at transplant date and at 6 months post-transplant) (Appendix A), and we followed those patients until the censor date. A very clear separation in risk of developing PTLD was observed between those in the high-risk and low-risk groups.

### 3.5. Sensitivity Analyses in Derivation Cohort

Including only the 46 definite PTLD cases in the analyses, 24 PTLD events were in the high-risk group and 22 PTLD events in the low-risk group. The PTLD incidence rate was 3.53 per 1000 PYFU (95% CI 2.51 to 4.55), and individuals in the high-risk group had a five times higher risk of definite PTLD compared with the low-risk group (IRR 5.28, 95% CI 2.96 to 9.39), and the discriminatory ability of the model was comparable to the main analysis (Harrell’s C-statistic 0.78, 95% CI 0.71 to 0.85). In sensitivity analysis, considering only the first year after transplant, there were 17 PTLD events, 15 in the high-risk group and 2 in the low-risk group (IRR 30.36, 95% CI 6.93 to 132.98). Harrell’s C-statistic for this model was 0.93 (95% CI 0.88 to 0.98). In comparison there were 40 PTLD events (18 in the high-risk group and 22 in the low-risk group) when only follow-up after the first year of transplant was considered. A lower but still significant risk of PTLD was observed in the high-risk group (IRR 4.03, 95% CI 2.17 to 7.48) and Harrell’s C-statistic was 0.71 (95% CI 0.63 to 0.80).

Both the model excluding the unknown values for the laboratory parameters and the model keeping them had very similar results. Retaining age group in the model led to a model with slightly poorer discriminatory ability, with a Harrell’s C-statistic of 0.81 (95% CI 0.71 to 0.91).

## 4. Discussion

The aim of this study was to develop a clinically useful risk score for predicting PTLD in solid organ transplant recipients and validate the score in a clinically diverse cohort. Our study included 2546 SOT recipients with 57 PTLD events (46 definite) in the Danish derivation cohort and 1611 SOT recipients with 24 definite PTLD events in the Swiss validation cohort. The risk score had a good discriminatory ability in both cohorts but seems to be most effective for the first year after transplant as suggested by the better discriminatory ability measured by higher Harrell’s C-statistic. Furthermore, the model was able to predict the risk of PTLD cases in both EBV positive and EBV negative individuals.

The risk score included characteristics consistently shown to impact the development of PTLD [6,7,20,21,22,23]. Although many studies have found that both sex and age were important risk factors for the development of PTLD [24,25,26], the inclusion of those two characteristics did not improve the model to predict PTLD in our study. On the other hand, the inclusion of transplant type, although non-significant in the univariate analysis, improved the model and was kept.

Below normal current hemoglobin levels were also associated with an increased risk of PTLD, and this was confirmed by the literature [27,28,29]. This has been reported in relation to bone marrow infiltration and immune hemolytic anemia [27] and after gastrointestinal blood loss due to PTLD in the bowels [28].

Although the risk score model performed well with good discriminatory ability in the entire follow-up period, the model performed even better during the first post-transplant year as seen through the C-statistics. The frequency of EBV DNA screening is higher in the first year after transplant where the patients are more intensively monitored, which could explain the better performance in the early post-transplant period. Another explanation could be that early and late occurring PTLD may be distinct entities with varying characteristics as suggested by previous studies [24,30]. For example, it has been shown that EBV DNA is less associated with late and more closely related to early PTLD. As EBV DNA is one of the strongest parameters in the risk score, this could explain why the risk score performs better in the early period. Nevertheless, excluding the first year after the transplant, the performance of the risk score was still good, and significant differences in the risk of PTLD were observed between the low- and high-risk groups.

In this study, if a patient has a positive EBV DNA alone (42 points), they will automatically fall into the high-risk group (score > 17 points) (Table 3). On the other hand, patients with negative or unknown EBV could still fall into the high-risk group according to the other components of the risk score. Thus, a strength of this score is the ability to predict potential future PTLD irrespective of whether the disease is related to EBV DNAemia.

The score has the potential to guide clinicians to decide which patients could benefit from more intensive monitoring for PTLD development. A recommendation would be to use the score more intensively in the high-risk group (e.g., every week or every other week) and then perform imaging if PTLD is suspected based on an increasing score, besides the standard clinical monitoring that is already part of the post-transplant follow-up. An additional suggestion is to use the score in a trial to identify high-risk individuals that may benefit from an early intervention.

The limitations of the study should be highlighted. EBV DNA measurements covered both those performed as part of the screening protocol and as part of a diagnostic procedure and therefore can be different between patient groups and over calendar time, as previously explained elsewhere [6]. However, all EBV DNA results were lagged to 28 days to reduce the impact of reverse causality [6]. Additionally, we observed that regular EBV DNA testing was carried out even in patients who were not formally screened, although less frequently. For the derivation cohort, around 30% of the patients had never been tested for EBV DNA (unknown category). However, all patients in the unknown category had a transplant date before 2010 (before the formal establishment of the MATCH cohort). The score assigned to the unknown EBV DNA category should be interpreted cautiously as the choice to measure EBV DNA or not is likely an artifact (confounding by indication) of those patients the physician deems to be at risk for PTLD. Therefore, in this study, it is expected that those patients with unknown EBV DNA present few risk factors for the development of PTLD and few clinical issues in general; thus, they were not administered an EBV DNA test. An ideal situation is to regularly request EBV DNA tests for all post-transplant patients.

The classification of PTLD in the two cohorts differed; the derivation cohort included definite, probable, and possible cases, while the validation cohort included only definite PTLD. However, sensitivity analyses including only definite PTLD for the derivation cohort showed similar results in relation to the discriminatory ability measured through Harrell’s C-statistic. In relation to the classification of lymphomas, the score also performed well for both cohorts, with more advanced lymphomas (validation cohort) and less advanced ones (derivation cohort).

Unfortunately, as mentioned before, data on immunosuppressive therapies (including rituximab) were not available at the individual patient level for this study, and the addition of this information could help to further improve the discriminatory ability of the risk score we have developed. According to clinicians of the validation cohort, their current immunosuppressive scheme did not differ more from the previous ones. For the kidney recipients in the derivation cohort, since 2004 the quadruple therapy is in use as “standard therapy”, with an induction phase and three different immunosuppressive drugs as a maintenance phase. The differences between the current immunosuppressive scheme and the previous ones were that for the induction phase daclizumab was used before 2008, and for the maintenance phase cyclosporine was used before 2012 and prednisolone since 2004, initially with a higher dose that was reduced in 2008 and with a further reduction from 2012 to minimize its possible side effects. Since the score performs well despite not being able to include immunosuppressive data, we expect that future inclusion of this parameter would most likely improve its performance. In addition, reliable information on rejection was not available for either cohort and could not be included the risk score.

## 5. Conclusions

In conclusion, a reliable risk score that predicts the risk of PTLD was developed and externally validated. This risk score is easy to calculate and could be used in future studies in a prospective matter and as part of the clinical routine for the screening of SOT recipients to determine the risk of developing PTLD, allowing more personalized monitoring strategies based on the patient’s individual risk such as more frequently imaging modalities of patients in the high-risk group for early detection and treatment of those with initial signs of emerging PTLD.

## Figures and Tables

**Figure 1 cancers-14-03279-f001:**
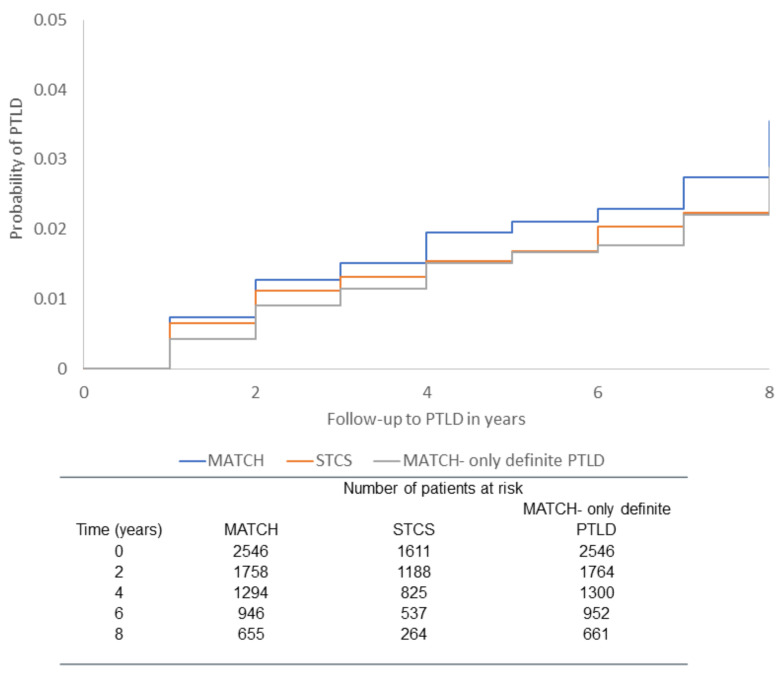
Kaplan–Meier with probability of PTLD for MATCH cohort; STCS cohort; MATCH with only definite PTLD.

**Table 1 cancers-14-03279-t001:** Characteristics of derivation and validation cohort at the time of transplant.

	MATCH (Derivation)						STCS (Validation)					
	All		Did Not Develop PTLD		Developed PTLD		All		Did Not Develop PTLD		Developed PTLD	
Characteristics	N or Median	% or IQR	N or Median	% or IQR	N or Median	% or IQR	N or Median	% or IQR	N or Median	% or IQR	N or Median	% or IQR
All	2546	100	2489	97.8	57	2.2	1611	100%	1587	98.5	24	1.5
Total follow-up (PYFU)	13,026.45		12,810.67		215.78		7218.53		7168.42		50.11	
Gender												
Male	1507	59.2	1473	59.5	34	59.6	1010	62.7	994	62.6	16	66.6
Female	1039	40.8	1016	41.5	23	40.4	601	37.3	593	37.4	8	33.4
Age group												
≤16 years	156	6.1	143	5.7	13	23.0	72	4.5	71	4.5	1	4.2
>16 years	2390	93.9	2346	94.3	44	77.0	1539	95.5	1516	95.5	24	95.8
Year of transplant	2012	(2008–2016)	2012	(2008–2016)	2010	(2007–2012)	2013	(2011–2016)	2013	(2011–2016)	2011	(2010–2014)
Type of transplant												
Heart	204	8.0	199	8.1	5	8.8	120	7.5	119	7.5	1	4.2
Lung	458	18.0	447	17.9	11	19.3	250	15.5	241	15.3	9	37.6
Kidney and/or pancreas ^1^	1201	47.2	1175	47.2	26	45.6	838	52.0	831	52.3	7	29.1
Liver	683	26.8	668	26.8	15	26.3	403	25.0	396	24.9	7	29.1
Donor/recipient EBV risk												
Low risk	2467	96.9	2418	97.1	49	86.0	1499	93.0	1480	93.2	19	79.2
High risk	79	3.1	71	2.9	8	14.0	112	7.0	107	6.8	5	20.8
Lab Tests												
Hemoglobin mmol/L	6.0	(5.4–6.6)	6.0	(5.4–6.6)	6.0	(5.4–6.5)	5.4	(4.8–6.1)	5.4	(4.8–6.1)	5.2	(4.8–5.7)
C-reactive protein mg/L	12.0	(5.0–32.0)	12.0	(5.0–32.0)	23.0	(6.0–59.0)	21.0	(8.0–48.0)	21.0	(8.0–48.0)	30.5	(13.5–52.5)
Platelets × 10^9^/L	203.0	(138.0–266.0)	203.0	(138.0–266.0)	188.0	(82.0–283.0)	187.0	(117.0–263.0)	187.0	(117.0–263.0)	198.5	(103.0–277.0)

^1^ 1174 had a kidney transplant, 25 patients combined pancreas + kidney transplant, and 2 single pancreas transplants.

**Table 2 cancers-14-03279-t002:** Characteristics of PTLD in the derivation and validation cohorts.

PTLD Characteristics	Derivation Cohort	Validation Cohort
Lymphocyte lineage: N (%)	B cell: 32 (56.1) Non-B cell: 14 (24.6) Missing ^a^: 11 (19.3)	B cell: 13 (54.2) Non-B cell: 11 (45.8)
Ann Arbor classification of Lymphoma: N (%)	Stage I: 13 (22.8) Stage II: 9 (15.8) Stage III: 5 (8.8) Stage IV: 13 (22.8) NA ^b^: 17 (29.8)	Stage I: 2 (8.3) Stage II: 2 (8.3) Stage III: 0 (0) Stage IV: 20 (83.4)
EBV DNA at time of PTLD diagnosis: N (%)	Negative: 6 (10.6) Positive: 28 (49.1) Missing: 23 (40.3)	Negative: 5 (20.8) Positive: 18 (75.0) Missing: 1 (4.2)

^a^ Patients with missing information are non-definite PTLD cases because they did not have a biopsy, and therefore it was not possible to assess lymphocyte lineage. ^b^ NA = not available; 11 out of 17 were not available because they were non-definite PTLD cases (no biopsy); 6 out of 17 did not have access to imaging.

**Table 3 cancers-14-03279-t003:** Median time and interquartile range (in days) between laboratory parameters measures in both cohorts.

	Periodicity in MATCH ^2^		Periodicity in STCS ^3^	
Laboratory Parameters	First Year after Transplant Median and IQR	Follow-up after First Year Median and IQR	First Year after Transplant Median and IQR	Follow-Up after the First Year Median and IQR
EBV DNA	16 days (7–44)	54 days (14–196)	8 days (6–14)	18 days (7–35)
CRP	2 days (1–10)	7 days (1–39)	1 day (1–6)	7 days (1–29)
Hemoglobin	3 days (1–13)	8 days (1–36)	1 day (1–6)	7 days (1–33)
Platelets	2 days (1–13)	10 days (1–45)	1 day (1–6)	7 days (1–33)

^2^ EBV DNA results were not available (not measured) for 789 patients and CRP; hemoglobin and platelets results were not available for 132 patients in the MATCH cohort. All those patients with missing values for laboratory parameters were transplanted before 2010 (before the establishment of the MATCH cohort). ^3^ EBV DNA results were not available (not measured) for 621 patients, and CRP, hemoglobin, and platelets results were not available for 40 patients in the STCS cohort.

**Table 4 cancers-14-03279-t004:** Multivariable model for the PTLD score in the derivation cohort (n = 2546).

Parameter ^a^	N of PTLD/Total PYFU	IRR WITH 95% CI	Exact Coefficient	Coefficient Used in Score Calculation
Intercept ^b^			−12.529	
Variables at baseline:				
Donor/recipient serostatus risk				
Low donor/recipient risk	49/12,676.98	1	0 (ref)	0
High donor/recipient risk	8/291.31	3.58 (1.32; 9.64)	1.275	19
Type of transplant				
Kidney or and pancreas transplant	26/7177.53	1	0 (ref)	0
Heart transplant	5/1133.55	1.81 (0.69; 4.68)	0.593	9
Liver transplant	15/2879.61	1.33 (0.68; 2.60)	0.286	4
Lung transplant	11/1777.59	1.06 (0.47; 2.38)	0.066	1
Time-updated variables:				
Hemoglobin levels (mmol/L) ^c^				
Hemoglobin§ < normal ranges ^d^	33/4826.75	1.98 (1.05; 3.71)	0.683	10
Hemoglobin normal	15/6167.38	1	0 (ref)	0
Hemoglobin > normal or missing	9/1974.16	2.12 (0.74; 6.02)	0.753	11
C-reactive protein levels (mg/L) ^c^				
C-reactive protein (<10 mg/L) or missing	37/10,934.90	1	0 (ref)	0
C-reactive protein ≥ 10 mg/L	20/2033.39	2.39 (1.35; 4.23)	0.873	13
Platelets levels (×10^9^/L) ^c^				
Platelets < normal or missing	15/2802.98	1.43 (0.64; 3.17)	0.360	5
Platelets normal	35/9554.03	1	0 (ref)	0
Platelets > normal	7/611.27	1.77 (0.78; 3.99)	0.571	9
EBV DNA categories ^c^				
Negative EBV DNA	25/7419.02	1	0 (ref)	0
EBV DNA-missing	13/5304.36	0.69 (0.34; 1.38)	−0.364	−6
Positive EBV DNA	19/244.91	16.34 (8.07; 33.10)	2.794	42

^a^ For each risk factor, only one level contributes to a patient’s risk of PTLD. For example, a high D/R patient gets 19 points for this variable. PTLD = post-transplant lymphoproliferative disorder; IRR = incidence rate ratio; CRP = C-reactive protein; EBV = Epstein–Barr virus. ^b^ Needed if exact risk is to be calculated. ^c^ Time-updated variables. ^d^ The reference range differs based on sex and age.

**Table 5 cancers-14-03279-t005:** Performance of the PTLD score in the derivation and validation cohorts (low-risk group: score ≤ 17 points; high-risk group: score > 17 points).

Variable	Derivation Cohort	Validation Cohort
Developed PTLD/no.	57/2546	24/1611
Incidence of PTLD per 1000 PYFU (95% CI)	4.40 (3.25–5.54)	3.32 (1.99–4.65)
Total follow-up (PYFU)	13,026.45	7218.53
Risk score model		
Score, median (IQR)	10.0 (0–14.0)	10.0 (1–16.0)
Score for those who developed PTLD, median (IQR)	20.0 (10.0–55.0)	22.0 (14.0–62.0)
N of PTLD by risk score group, low/high risk	24/33	8/16
N of PYFU by risk score group, low/high risk	10,776.14/2192.17	5644.28/1574.26
Incidence of PTLD per 1000 PYFU (95% CI)		
Low risk (score ≤ 17)	2.23 (1.34–3.12)	1.42 (0.44–2.40)
High risk (score > 17)	15.05 (9.92–20.19)	10.16 (5.18–15.14)
Incidence rate ratio (95% CI)		
Low risk (score ≤ 17)	1 (ref)	1 (ref)
High risk (score > 17)	6.75 (4.00–11.41)	7.17 (3.05–16.82)
Incidence rate ratio per 1-point increase in score	1.06 (1.05–1.08)	1.06 (1.04–1.08)
Harrell’s C-statistic	0.82 (0.76–0.88)	0.82 (0.72–0.92)
Brier score	0.000084	0.000064

## Data Availability

The datasets generated and/or analyzed during the current study are available from the corresponding author on reasonable request.

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
