# Peer review of "Development and Validation of a Risk Score for Post-Transplant Lymphoproliferative Disorders among Solid Organ Transplant Recipients"

_cancers, 2022, doi:10.3390/cancers14133279_

Round 1
Reviewer 1 Report
I have found this a very informative paper about the development and validation of a risk score for post-transplant lymphoproliferative disorders among solid organ transplant recipients. I has been developed using both a huge derivation and validation cohort of solid organ transplants. The develop model seems to performs particularly well predicting EBV PTLD in the first year after transplant. But less well predicting PTLD after the first year.
Some questions remain
1) Are there data on this patients on the amount of immunosupression/ steroid use and how does that predict PTLD. Or maybe symptoms of rejection can be used as a biomarker for extra immunosuppressive treatment to assess that.
2) Are there data on immune status ( CD4/ CD8 T cells, NKcells, B cells) of these patients?
3) Were patients treated with rituximab in response to EBV DNAemia and does pre emptive treatment influence the risk of development of PTLD. Does rituximab treatment interfere with how well your score performs?
4) Predictors are only helpful if they warn well enough in advance. To me it is unclear at which timepoint the score ( and thus highest(?) EBV DNAemia and CRP timing) were used to try and predict in both cohorts .
Author Response
Reviewer 1
Comments and Suggestions for Authors
I have found this a very informative paper about the development and validation of a risk score for post-transplant lymphoproliferative disorders among solid organ transplant recipients. It has been developed using both a huge derivation and validation cohort of solid organ transplants. The develop model seems to perform particularly well predicting EBV PTLD in the first year after transplant. But less well predicting PTLD after the first year.
Some questions remain:
- Are there data on this patients on the amount of immunosuppression/ steroid use and how does that predict PTLD. Or maybe symptoms of rejection can be used as a biomarker for extra immunosuppressive treatment to assess that.
R: Thank you for the comment. As mentioned in the paper (page 3, lines 104-107 and page 13, lines 172-174 and lines 180-183 – see extracts below), unfortunately we do not have data on the individual amount of immunosuppression/ steroid use. We also added a comment on the performance of the score in the future in case this variable is available. In addition, information on rejection for this population is not reliable for both cohorts (we included this information on page 13, lines 175-176)
“Individual patient data on specific immunosuppressive therapies was not available however, detailed information on the immunosuppressive schemes for both cohorts can be found in previous published articles (14, 15) and as a Supplemental material.”
“Unfortunately, as mentioned before, data on immunosuppressive therapies were not available at the individual patient level for this study and the addition of this information could help to further improve the discriminatory ability of the risk score we have developed…
“Since the score performs well despite not being able to include this confounder, we expect that future inclusion of this parameter would most likely improve its performance.”
New information included about rejection:
“In addition, reliable information on rejection was not available for either cohort and could not be included in the risk-score.”
- Are there data on immune status (CD4/ CD8, T cells, NK cells, B cells) of these patients?
R: Unfortunately, we do not have data on immune status. However, we did previously assess the risk score also including leukocyte count during follow-up, which did not improve the score.
3) Were patients treated with rituximab in response to EBV DNAemia and does pre emptive treatment influence the risk of development of PTLD. Does rituximab treatment interfere with how well your score performs?
R: Thank you for the comment. As mentioned in our response to question 1, unfortunately we do not have reliable data on individual use of medications including rituximab. For the MATCH cohort, individuals who received rituximab prior to PTLD development would most likely have been included as probable or possible PTLD as this treatment would generally be administered in patients with early clinical signs of PTLD. Furthermore, if one imagines that the treatment with rituximab could prevent the occurrence of PTLD, then it could be argued that by not including it we would underestimate the risk. Since the score performs well despite not being able to include this confounder, we expect that future inclusion of this parameter would most likely improve the performance of the risk-score.
We have added the following to the discussion to clarify this (page 13, lines 172-174):
“Unfortunately, as mentioned before, data on immunosuppressive therapies (including rituximab) were not available at the individual patient level for this study and the addition of this information could help to further improve the discriminatory ability of the risk score we have developed.”
4) Predictors are only helpful if they warn well enough in advance. To me it is unclear at which timepoint the score (and thus highest(?) EBV DNAemia and CRP timing) were used to try and predict in both cohorts.
R: The score can be used at any time point after transplant (page 4, lines 174-178). We emphasize that the only thing needed is to enter the patient’s characteristics available at a certain time point (line 176) In the paper, we gave an example where we predicted the risk of developing PTLD in the next 180 days using the last available information on patients’ parameters (page 4, line 179 and page 11 lines 78-79).
The risk score proposed in this study can be used at any time point after the transplant. If for example, we want to calculate the risk of developing PTLD in the next 180 days it can be calculated using the formula:
Using the exact coefficients, this same patient would have a risk for developing PTLD in the next 180 days of 8.73%.

Reviewer 2 Report
The authors have developed a stratification risk tool to identify SOT patients with an increased risk of developing PTLD. The effort is laudable given the difficulties in identifying the patients with PTLD and the rarity of PTLD. Moreover, there are no specific guidelines for the screening. The authors used 2 large cohorts of patients to validate the risk score. Overall, the incidence of PTLD in their 2 cohorts is only 2% (57 over 2456 pts) and 1.4% (24 over 1611 pts) which is in line with what was reported in the literature.
The manuscript represents a helpful effort to develop a clinically useful tool to discriminate the risk of developing PTLD. The risk score appears to perform better in the patients who are known to carry a higher risk of developing PTLD (EBV mismatch and positive EBV DNA). It would be interesting to use this risk tool in further studies in a prospective matter.
MINOR EDITS
Page 4, Linme 154: " time" is in a larger font and bold. correct
page 8:
Table 2:
- some parts are in Bold other are not. Please correct
- Include a space between 2 and EBV DNA results
Table 3 exchanges superscript numbers with letters (i.e. Parameter4 to Parametera)
Page 10, line 66, delete a negative EBV DNA (+0).
Author Response
Reviewer 2
The authors have developed a stratification risk tool to identify SOT patients with an increased risk of developing PTLD. The effort is laudable given the difficulties in identifying the patients with PTLD and the rarity of PTLD. Moreover, there are no specific guidelines for the screening. The authors used 2 large cohorts of patients to validate the risk score. Overall, the incidence of PTLD in their 2 cohorts is only 2% (57 over 2456 pts) and 1.4% (24 over 1611 pts) which is in line with what was reported in the literature.
The manuscript represents a helpful effort to develop a clinically useful tool to discriminate the risk of developing PTLD. The risk score appears to perform better in the patients who are known to carry a higher risk of developing PTLD (EBV mismatch and positive EBV DNA). It would be interesting to use this risk tool in further studies in a prospective matter.
R: Thank you very much for the comment. We have included a sentence with the potential use of this tool in prospective studies as well (page 13, lines 186 and 187).
“This risk score is easy to calculate and could be used in future studies in a prospective matter and as part of the clinical routine for the screening of SOT…”
MINOR EDITS
Page 4, Line 154: " time" is in a larger font and bold. correct
R: Thank you for your comment. We correct it in the current version.
page 8:
Table 2:
- some parts are in Bold other are not. Please correct
R: It is corrected in the current version.
- Include a space between 2 and EBV DNA results
R: We included a space in the current version.
Table 3 exchanges superscript numbers with letters (i.e. Parameter4 to Parametera)
R: We corrected it in the current version.
Page 10, line 66, delete a negative EBV DNA (+0).
R: We deleted it in the current version.

Reviewer 3 Report
In this manuscript, dos Santos et al set out to generate and test a risk score for predicting the development of post-transplant lymphoproliferative disease (PTLD). The authors develop a risk score in their own centre (mostly relating to EBV serostatus, EB viraemia, type of transplant (presumably as a proxy measure of immunosuppression intensity) and a number of biochemical/haematological parameters. This risk score is then validated in a second, geographically distinct cohort. In both cases, the score provides good predictive power with a c-statistic of over 0.8.
The issue of PTLD is a common one amongst transplant recipients and is one of the more prevalent malignancies - as such, a method to stratify patients by risk is sorely needed.
The authors should be congratulated on writing a strong manuscript and pulling together two large cohorts to do this. However, I do have some comments which I feel need addressing prior to acceptance.
Major:
1. Throughout the manuscript the authors refer to PTLD as if it were a single disease entity - but as mentioned in their introduction this is a somewhat heterogenous condition with a spectrum of presentations. Greater detail on the presentation of PTLD included in the development and validation cohorts is therefore needed. Examples include stage at diagnosis, lymphocyte lineage, histological appearance, etc.
2. Much of the weighting on the risk score relates to EBV - by either being EBV D+R- in itself, or additional EBV viraemia - will lead to a high-risk stratification. EBV is generally related to B cell lymphoma and it may be that this score is particularly useful in predicting this subtype. However the authors suggest that this score is still useful in predicting EBV-negative PTLD. Around 30% of PTLD will be of a T or NK cell lineage - and is therefore unrelated to EBV. Could the authors provide separate ROC curves for B-cell and non-B cell PTLD and those who are EBV positive and negative on histology?
3. It would be helpful to provide positive and negative predictive values for PTLD development in high-risk patients.
4. Kaplan-Meier curves for development of PTLD stratified by risk score might be useful to help visualise the development over time.
5. Lack of information about immunosuppression as a limitation should also mention the widespread change in immunosuppression over time - especially in renal transplantation following the publication of the ELITE-SYMPHONY trial, leading to a shift away from ciclosporin/azathioprine and towards tacrolimus/mycophenolate. It may be helpful, in the absence of specific data regarding this in the cohort, to discuss changes in both the derivation and validation units' practices over this time.
Minor comments:
1. Table 1 should also include duration of follow-up for participants.
2. Page 10 line 66 demonstrates repetition ("a negative EBV DNA")
Author Response
Reviewer 3
Comments and Suggestions for Authors
In this manuscript, dos Santos et al set out to generate and test a risk score for predicting the development of post-transplant lymphoproliferative disease (PTLD). The authors develop a risk score in their own centre (mostly relating to EBV serostatus, EB viraemia, type of transplant (presumably as a proxy measure of immunosuppression intensity) and a number of biochemical/haematological parameters. This risk score is then validated in a second, geographically distinct cohort. In both cases, the score provides good predictive power with a c-statistic of over 0.8.
The issue of PTLD is a common one amongst transplant recipients and is one of the more prevalent malignancies - as such, a method to stratify patients by risk is sorely needed.
The authors should be congratulated on writing a strong manuscript and pulling together two large cohorts to do this. However, I do have some comments which I feel need addressing prior to acceptance.
Major:
- Throughout the manuscript the authors refer to PTLD as if it were a single disease entity - but as mentioned in their introduction this is a somewhat heterogenous condition with a spectrum of presentations. Greater detail on the presentation of PTLD included in the development and validation cohorts is therefore needed. Examples include stage at diagnosis, lymphocyte lineage, histological appearance, etc.
R: Thank you very much for your comment. We have included a new table in the paper (Table 2, page 8) with additional information related to the PTLD cases and information located in page 8, lines 2-7 (see below).
Characteristics of PTLD for both cohorts are found in Table 2. A little over half of the PTLD cases in both cohorts were of B cell origin (56.1% and 54.2% in derivation and validation cohorts respectively). In relation to Ann Arbor classification of Lymphoma, for the validation cohort almost all of the PTLDs were classified as stage IV (most advanced stage of lymphoma), while for derivation cohort, there was a more even distribution across the stages. Further, in both cohorts the majority of definite PTLD cases were positive for EBV DNA in the biopsies at time of PTLD diagnosis.
We have also highlighted tin the discussion that the score performs well with both more advanced (validation cohort) and less advance (derivation cohort) lymphomas (page 13, lines 169-171).
In relation to the classification of lymphomas, the score also performed well for both cohorts, with more advanced lymphomas (validation cohort) and less advanced ones (derivation cohort).
- Much of the weighting on the risk score relates to EBV - by either being EBV D+R- in itself, or additional EBV viraemia - will lead to a high-risk stratification. EBV is generally related to B cell lymphoma and it may be that this score is particularly useful in predicting this subtype. However, the authors suggest that this score is still useful in predicting EBV-negative PTLD. Around 30% of PTLD will be of a T or NK cell lineage - and is therefore unrelated to EBV. Could the authors provide separate ROC curves for B-cell and non-B cell PTLD and those who are EBV positive and negative on histology?
R: Thank you for your comment. We did not include the ROC curves for B-cell vs non-B cell in our paper since the number of PTLDs was already very low in both cohorts (especially in the validation cohort- only 24 PTLDs- and if we further stratify this category into B-cell and non-B cell we would have even lower numbers- therefore we would lose a lot of power in our analyses or obtain biased results). The same reason prevented us from generating ROC curves for positive vs negative EBV at the time of PTLD diagnosis. However, we will consider this in the future if validating the score in a larger cohort.
- It would be helpful to provide positive and negative predictive values for PTLD development in high-risk patients.
R: Many thanks for your suggestion. Our model is not a static model, because we used time-updated analysis to account for each new available laboratory measurements of the patients. Thus, patients can change groups all the time and it is not possible to determine an overall number of patients in the high-risk group that did not develop PTLD.
However, in the Kaplan-curves, which were suggested in the next comment, when we look at the risk over-time from a fixed time-point post-transplant (at transplant and at 6 months post-transplant) and follow those patients until the censor date we see a very nice separation in risk between those in the high risk at low-risk groups at 6 months. See supplemental material 2 and on page 11, lines 84-87.
“In relation to the probability of PTLD, we fixed two specific time-points (at transplant date and at 6 months post-transplant) (supplemental material 2) and we followed those patients until the censor date. It was observed a very clear separation in risk of developing PTLD between those in the high risk and low-risk groups.”
- Kaplan-Meier curves for development of PTLD stratified by risk score might be useful to help visualise the development over time.
R: Thank you very much for your comment. We now included the Kaplan Meier curves (as explained in the previous comment) as a supplemental material (Supplemental material 2). One curve with the risk score evaluated at the time of transplant and the other with the score evaluated at 6 months in those still under follow-up.
- Lack of information about immunosuppression as a limitation should also mention the widespread change in immunosuppression over time - especially in renal transplantation following the publication of the ELITE-SYMPHONY trial, leading to a shift away from ciclosporin/azathioprine and towards tacrolimus/mycophenolate. It may be helpful, in the absence of specific data regarding this in the cohort, to discuss changes in both the derivation and validation units' practices over this time.
R: Thank you very much for your comment. Recent studies have shown that azathioprine related regimens result in higher risk of PTLD. Therefore, one could imagine that shift in immunosuppressive regimens over calendar period could impact PTLD incidence. However, even though fewer toxic regimens may be used in newer time, this is not translated into lower incidence of PTLD over time. This is likely due to more awareness and that the transplant population is older and have higher risk of cancer in general, including lymphomas. For this study, according to the clinicians of the validation cohort (STCS), the current immunosuppressive scheme (available in reference 15 of the paper) does not differ much from the previous ones. For the derivation cohort, the current immunosuppressive scheme is seen in both reference 14 and in the Supplemental material 1. For kidney recipients, since 2004 the quadruple therapy is in use as “standard therapy”, meaning an induction phase+ 3 different immunosuppressive drugs as maintenance phase. The differences from the current immunosuppressive scheme to the previous ones were that for the induction phase Daclizumab was used before 2008 and for the maintenance phase Cyclosporine was used before 2012 and prednisolone is in use since 2004, initially with a higher dose that was reduced in 2008 and with a further reduction from 2012, to reduce its possible side effects. We included this information in the text, on page 13, lines 174-180.
According to clinicians of the validation cohort, their current immunosuppressive scheme did not differ more from the previous ones. For the kidney recipients in the derivation cohort, since 2004 the quadruple therapy is in use as “standard therapy”, with an induction phase and three different immunosuppressive drugs as maintenance phase. The differences from the current immunosuppressive scheme to the previous ones were that for the induction phase Daclizumab was used before 2008 and for the maintenance phase Cyclosporine was used before 2012 and prednisolone is in use since 2004, initially with a higher dose that was reduced in 2008 and with a further reduction from 2012 to minimize its possible side effects
Minor comments:
- Table 1 should also include duration of follow-up for participants.
R: Thank you for the comment. We included the follow-up for participants in Table 1.
- Page 10 line 66 demonstrates repetition ("a negative EBV DNA")
R: Thank you for noticing my mistake. I deleted the repetition.

Round 2
Reviewer 3 Report
Thank you for incorporating the changes made. I have no further concerns or comments and applaud the authors on this piece of work.